# Severe drought exposure in utero associates to children's epigenetic age acceleration in a global climate change hot spot

Xi Qiao [1,8], Bilinda Straight [2,8] ✉, Duy Ngo[1,8], Charles E. Hilton[3], Charles Owuor Olungah[4], Amy Naugle[5], Claudia Lalancette [6] & Belinda L. Needham[7,8]

The goal of this study is to examine the association between in utero drought exposure and epigenetic age acceleration (EAA) in a global climate change hot spot. Calculations of EAA in adults using DNA methylation have been found to accurately predict chronic disease and longevity. However, fewer studies have examined EAA in children, and drought exposure in utero has not been investigated. Additionally, studies of EAA in low-income countries with diverse populations are rare. We assess EAA using epigenetic clocks and two DNAm-based pace-of-aging measurements from whole saliva samples in 104 drought-exposed children and 109 same-sex sibling controls in northern Kenya. We find a positive association between in utero drought exposure and EAA in two epigenetic clocks (Hannum's and GrimAge) and a negative association in the DNAm based telomere length (DNAmTL) clock. The combined impact of drought's multiple deleterious stressors may reduce overall life expectancy through accelerated epigenetic aging.

Climate change will have substantial consequences for human health. Moreover, a disproportionate impact is predicted in communities engaging in climate-sensitive livelihoods such as pastoralism. This is particularly true in global hot spots such as East Africa and for vulnerable groups such as pregnant individuals, young children, and the elderly. Adapting to climate change requires enhanced understanding of factors for resilience versus vulnerability, and their biological as well as behavioral mechanisms. Numerous studies have linked in utero exposure to severe drought to adverse pregnancy outcomes and alterations in offspring growth and body composition[1–4]. However, drought has multiple, simultaneous impacts on the mother, such as exacerbated food and water insecurity (often with accompanying dehydration), and psychosocial stress[5,6]. Although there is some limited evidence for biological mechanisms for individual drought-related stressors, particularly in animal studies[7,8], the biological mechanisms

and lifelong implications of drought's combined impact for child outcomes are not well understood[9].

Accelerated molecular aging as measured through epigenetic "clocks" is emerging as a potential mechanism whereby psychosocially and physiologically stressful conditions lead to adverse health conditions and mortality[10]. Yet there are no human studies to date that measure associations between epigenetic aging and early life exposure to the climate change impacts of severe drought. Moreover, there are scant studies of epigenetic aging in low-income countries with ethnically and racially diverse populations[11]. In addition to the importance of diversity to understanding biological aging in humans globally, these populations are among the most vulnerable to the effects of climate change[12].

DNA methylation (DNAm) levels at numerous cytosine-phosphate-guanine sites (CpGs) have been found to be accurate

[1]Department of Statistics, Western Michigan University, Kalamazoo, MI, USA. [2]School of Environment, Geography, & Sustainability, Western Michigan University, Kalamazoo, MI, USA. [3]Department of Anthropology, University of North Carolina - Chapel Hill, Chapel Hill, NC, USA. [4]Department of Anthropology, Gender and African Studies, University of Nairobi, Nairobi, Kenya. [5]Department of Psychology, Western Michigan University, Kalamazoo, MI, USA. [6]Epigenomics Core, University of Michigan, Ann Arbor, MI, USA. [7]Department of Epidemiology, School of Public Health, University of Michigan, Ann Arbor, MI, USA. [8]These authors contributed equally: Xi Qiao, Bilinda Straight, Duy Ngo, Belinda L. Needham. ✉e-mail: bilinda.straight@wmich.edu

biomarkers of molecular aging, more promising than other biomarkers of biological aging[13]. Utilizing machine learning methods, a number of epigenetic clocks have been developed to predict biological age, lifespan, and mortality by regressing chronological age, health-related outcomes, or biomarkers on a set of CpGs, selecting the most informative ones[14].

Previous studies have shown that higher epigenetic age relative to chronological age, i.e., epigenetic age acceleration (EAA), predicts poor health outcomes, particularly metabolic syndrome and other chronic diseases, and earlier age at death[15,16]. Fewer studies assess EAA associations in children, although recent studies in children have linked EAA to a variety of exposures in utero, including metabolic disorders such as gestational diabetes mellitus, maternal tobacco smoking, and indoor particulate matter absorbance[17,18].

Our study employs a quasi-experimental same-sex sibling design to examine offspring outcomes of exposure in the first trimester of gestation in utero to a 2008–2009 severe drought in East Africa. Even with humanitarian support from the World Bank and the European Union, the drought brought devastating consequences to pastoralist

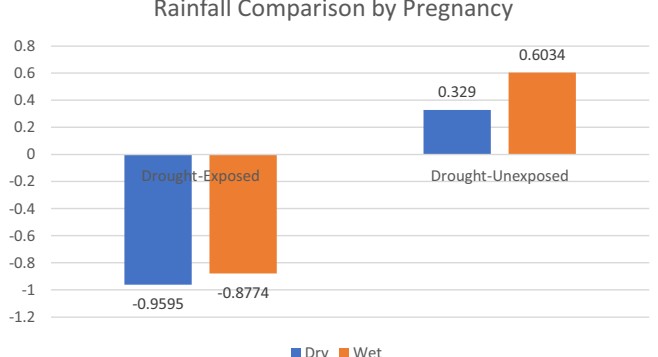

**Fig. 1 | Rainfall comparison by pregnancy.** Rainfall z-scores in comparison to 40-year mean for 3-month period leading to conception (indicative of pasture quality in first trimester) by pregnancy and mother's reported residence, using CHIRPS rainfall data. Figure shows mean z-scores by pregnancy and location for drought-exposed and unexposed pregnancies in dry (blue columns) and wet (orange columns) seasons. Source data are provided as a Source Data file.

communities in northern Kenya, including the Samburu, who are the focus of our study[19]. Drought monitoring experts consider the 2008–2009 drought to be severe as measured meteorologically, in terms of rangeland, need for food aid, and with respect to livestock losses – which were 57% for cattle and 65% for sheep[19]. Although our Samburu study partners report pregnancy as a stressful time overall, they report substantially more distress during pregnancies in the 2008–2009 drought compared to pregnancies with same-sex sibling controls conceived after the drought resolved and unexposed to severe drought in utero[20]. In Fig. 1, we draw on CHIRPS data[21] to provide rainfall z-scores for drought-exposed versus unexposed pregnancies based on the 40-year cumulative mean rainfall during the same period. The rainfall period is based on health records and participant reports of the location for each of their pregnancies.

We have previously reported that severe drought exposure in utero associated to lower child body weight and higher peripheral adiposity independently of microclimate variation[20]. Additionally, we have previously reported differential epigenome-wide DNAm patterns based on severe drought exposure, as well as potential CpG sites that mediate the association between drought exposure and child body weight and potentially peripheral body fat[22]. In the present study, we hypothesize an association between drought exposure in utero and EAA. We provide a conceptual diagram for our study in Fig. 2.

For assessing EAA, we use the most widely validated first generation clocks for consistency with other studies, including those in children. Horvath's clock is drawn from a variety of tissues from individuals representing a broad age range (0–100 years), and is accurate in young children[23], although racialized group membership for the training data was only partially reported (non-Hispanic White; Taiwanese)[24]. The similarly pan-tissue skin and blood clock (trained on ages 0 to 94 years) has been found to have even higher accuracy for age estimation than Horvath's clock[25]. Hannum's single-tissue clock might be less accurate in children for age estimation, although it is a more accurate predictor of life expectancy[14] and several studies have found significant associations in children based on early life adversity[26]. The training data set for Hannum's clock included 426 White and 230 Hispanic individuals, ages 19–101 years[27].

While first generation clocks were developed to predict chronological age, second generation clocks were trained specifically to

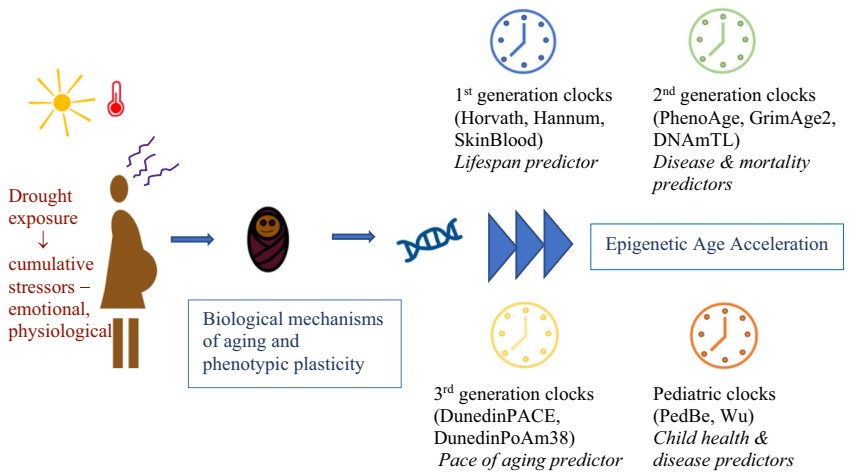

**Fig. 2 | Conceptual model for study.** Drought exposure during pregnancy indicated (gold sun and red thermometer weather symbols and brown diagram of pregnant woman) hypothesized to predict child outcomes (brown infant diagram) through biological mechanisms of aging and phenotypic plasticity (blue DNA symbol), tested in this study with epigenetic age acceleration (blue forward arrows) using multiple epigenetic clocks, as labeled in Figure as follows: 1st generation clocks (blue clock) – Horvath, Hannum, SkinBlood, identified as lifespan predictors; 2nd generation clocks (green clock) – PhenoAge, GrimAge2, DNAmTL, identified as disease and mortality predictors; 3rd generation pace of aging predictors (yellow clock) – DunedinPACE, DunedinPoAm38; and pediatric clocks (orange clock) – PedBe and Wu, identified as child health and disease predictors.

predict diseases and mortality. We use the PhenoAge[28] and GrimAge[29] clocks for their reported relative precision in predicting disease-related aging and mortality, as well as two recent clocks (DunedinPoAm38[30] and DunedinPACE[31]) that were developed to measure the pace of aging over time based on adult samples[32,33]. DunedinPACE provides refinements that increase its precision, while DunedinPoAm has been around longer and therefore featured in more studies. Racialized identities of participants for training data were not identified for GrimAge or PhenoAge, although, for GrimAge, stratified validation testing was performed based on racialized group membership (Black, White, Hispanic) and education levels. Dunedin was trained almost entirely on White New Zealanders of a broad range of socioeconomic backgrounds. We also test the DNAmTL, which was developed based on leukocyte telomere length in adults ages 22–93 years (19% European and 81% African ancestry) but is applicable to children. DNAmTL was validated in samples derived from participants identified as European, African, and Hispanic ancestries. Telomeres (the protective caps at the ends of chromosomes) shorten with each cell division, as well as in response to oxidative stress; shortened telomere length has been found to associate to psychosocial stress, age-related diseases, and mortality[34,35]. The DNAmTL reflects the replicative history of cells, negatively correlates with age in different tissues and cell types and is a strong predictor of mortality and multiple health outcomes[36].

Most clocks were trained on adults, or, in the case of Horvath's clock, on all ages. Since our study is in children, we include two pediatric clocks. The PedBE clock was developed from buccal epithelial cells (from predominantly White participants) for use specifically in children ages zero to twenty years[37]. A very recent pediatric clock, Wu's clock, was trained on children using blood samples of children ages 9–212 months of age. Training data included participants identified as White, Asian, African-American, and multiple racialized identities, although this information was only available for 2 out of 11 data sets. In validation tests, gender and ethnicity did not seem to influence epigenetic age acceleration. Based predominantly on CpG sites in genes relevant to development and aging, the clock was designed to predict age-related diseases at young ages to allow positive interventions[38].

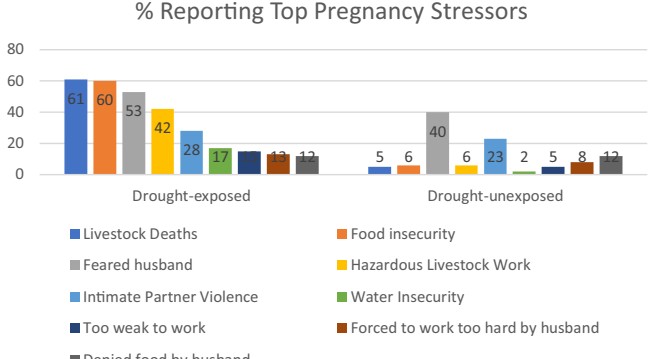

**Fig. 3 | Frequently reported pregnancy stressors by percentage reported.** Top stressors reported as experienced during drought pregnancy shown in comparison to reporting of those same stressors during drought-unexposed pregnancy ($N$ reporting = 104 Drought-exposed, $N$ reporting = 109 Drought-unexposed), based on percentage of women reporting the stressor by pregnancy. Stressors as indicated in figure columns are livestock deaths (blue), food insecurity (orange), feared husband (gray), hazardous livestock work (yellow), intimate partner violence (light blue), water insecurity (green), too weak to work (navy), forced to work too hard by husband during the pregnancy (brown), denied food by husband during pregnancy (husband refusing to provide food) (dark gray). Source data are provided as a Source Data file.

## Results

### Maternal exposures and descriptive statistics

This study aims to examine EAA in children exposed to severe drought in utero ($N = 104$) compared to same-sex siblings unexposed to severe drought in utero ($N = 109$). Figure 3 shows the top self-reported stressors reported for drought pregnancies. We have previously reported all stressors: there were significantly higher frequencies of most stressors reported for drought compared to typical season pregnancies, indicative of the cumulative, exacerbated stress characteristic of drought[22]. Descriptive statistics for variables included in the model are shown in Table 1. More girls than boys are seen in our study. Exhaustive descriptive statistics and additional cultural context for maternal stressors have been provided previously[20,22].

### Epigenetic clocks and epigenetic age acceleration

The average chronological age of our study sample is 6.72 years. Based on the same-sex sibling design, drought-exposed siblings are, on average, older than same-sex siblings conceived after the severe drought resolved. The means of the estimated epigenetic ages are 7.27 (PedBE), 6.25 (Wu), 8.39 (Horvath), 18.32 (Hannum), 4.02 (skin & blood), −3.32 (PhenoAge), 32.28 (GrimAge2), 7.19 (DNAmTL), and 4.02 (skin & blood). Age prediction accuracy is measured by the Pearson correlation (r) between the epigenetic clocks and child chronological age[39], and we found that all estimated biological ages from epigenetic clocks are correlated to child chronological age (Table 2, Fig. 4a), with the skin & blood clock showing the highest accuracy of age prediction ($r = 0.86$), PedBE the next ($r = 0.80$), Horvath ($r = 0.72$) and Wu ($r = 0.68$) slightly lower and DNAmTL the lowest ($r = 0.05$). Estimated epigenetic ages from different clocks are also correlated with each other (Fig. 4a), i.e., Horvath and skin & blood ($r = 0.79$), PedBE and skin & blood ($r = 0.78$), Wu and skin & blood ($r = 0.72$), Hannum and GrimAge2 ($r = 0.72$), Wu and Horvath ($r = 0.69$), and PedBE and Horvath ($r = 0.61$). DNAmTL is negatively correlated with five of the estimated epigenetic ages (Hannum ($r = 0.68$), GrimAge2($r = 0.67$), PhenoAge ($r = 0.35$), Wu ($r = −0.11$), PedBE ($r = −0.06$)). For DNAm based aging pace measurements, DunedinPACE negatively correlates with chronological age (mean = 1.32, $r = −0.35$,) while DunedinPoAm38 shows a weak positive correlation with age (mean = 1.16, $r = 0.18$). For EAA and aging pace measurements, the correlation between estimated EAA from GrimAge2 ($EAA_{GrimAge}$) and $EAA_{DunedinPACE}$ ($r = 0.74$), between $EAA_{Hannum}$ and $EAA_{GrimAge}$ ($r = 0.70$), between $EAA_{Hannum}$ and $EAA_{DunedinePACE}$ ($r = 0.63$), and between $EAA_{DNAmTL}$ and $EAA_{DunedinePoAm38}$ ($r = 0.52$) are higher than positive correlations between any other two EAA measures (Fig. 4b). EAA estimated from

**Table 1 | Descriptive statistics for drought exposure and covariates**

| | Mean (SD) | N (%) |
|---|---|---|
| Maternal exposure | | |
| Drought exposed | | 104/213 (49%) |
| Unexposed | | 109/213 (51%) |
| Child's sex | | |
| Female | | 115/213 (54%) |
| Male | | 98/213 (46%) |
| Gravida | 3.08 (2.21) | |
| Child birth season | | |
| Wet | | 126/213 (59%) |
| Dry | | 87/213 (41%) |
| Epithelial cell proportion[a] | 0.24 (0.12) | |
| Fibroblast cell proportion[a] | 0.02 (0.01) | |

[a]Cellular heterogeneity: Immune cells are reference. Source data are provided as a source data file.

**Table 2 | Descriptive statistics for chronological age, epigenetic clocks, aging pace measures, and age acceleration measures, with pearson correlation coefficients for associations with chronological age**

| | Mean | SD | Maximum | Minimum | Pearson correlation coefficients with chronological age |
|---|---|---|---|---|---|
| Child chronological age (years) | 6.72 | 1.96 | 9.61 | 1.81 | 1 |
| **Epigenetic clocks** | | | | | |
| PedBE | 7.25 | 1.15 | 10.16 | 3.99 | 0.80 |
| Wu | 6.25 | 1.66 | 10.17 | 0.51 | 0.68 |
| Horvath | 8.39 | 2.77 | 15.40 | 2.72 | 0.72 |
| Hannum | 18.32 | 3.62 | 31.02 | 8.71 | 0.38 |
| Skin & blood | 4.02 | 1.44 | 8.13 | 0.73 | 0.86 |
| PhenoAge | −3.32 | 6.49 | 17.39 | −21.04 | 0.51 |
| GrimAge2 | 32.28 | 3.93 | 48.24 | 23.66 | 0.25 |
| DNAmTL | 7.19 | 0.28 | 7.78 | 6.13 | 0.05 |
| **Aging pace measures** | | | | | |
| DunedinPACE | 1.32 | 0.14 | 1.78 | 1.01 | −0.35 |
| DunedinPoAm38 | 1.16 | 0.07 | 1.32 | 0.99 | 0.18 |
| **Age acceleration measures** | | | | | |
| EAA$^a_{PedBE}$ | 0.00 | 0.69 | 1.95 | −1.98 | 0 |
| EAA$_{Wu}$ | 0.00 | 1.21 | 2.67 | −3.39 | 0 |
| EAA$_{Horvath}$ | 0.00 | 1.92 | 5.59 | −3.89 | 0 |
| EAA$_{Hannum}$ | 0.00 | 3.35 | 11.50 | −7.48 | 0 |
| EAA$_{Skin\&Blood}$ | 0.00 | 0.73 | 2.62 | −2.14 | 0 |
| EAA$_{PhenoAge}$ | 0.00 | 0.69 | 1.96 | −1.98 | 0 |
| EAA$_{GrimAge2}$ | 0.00 | 1.77 | 5.14 | −3.58 | 0 |
| EAA$_{DNAmTL}$ | 0.00 | 3.35 | 11.50 | −7.48 | 0 |

$^a$EAA denotes the epigenetic age acceleration, which is calculated as the resulting residuals of regressing epigenetic clock on child's chronological age. For example, EAA$_{PedBE}$ is the epigenetic age acceleration derived by taking the residuals after regressing PedBE clock on child's chronological age. Source data are provided as a source data file.

DNAmTL clock is negatively correlated with the majority of the age pace and acceleration measures (Hannum ($r = −0.75$), GrimAge2 ($r = −0.71$), DunedinPACE ($r = −0.67$), PhenoAge ($r = −0.44$), Wu ($r = −0.19$), PedBE ($r = −0.16$)). As expected, none of the EAAs are associated with chronological age ($r = 0$).

**Association of severe drought exposure in utero with EAA**

The posterior mean and 95% highest posterior density (HPD) interval of the estimated association of maternal exposure to severe drought with EAAs and aging pace measurements from our multivariate linear mixed effect model (see Methods) are summarized in Table 3. Detailed estimated regression coefficients showing the association between EAA measures and maternal exposure to severe drought, child gender, and other covariates are shown in Supplementary Table 1. Drought exposure is positively associated with EAA using Hannum's clock (posterior mean = 1.34, 95% HPD = (0.74, 1.96)), and GrimAge2 (posterior mean = 1.31, 95% HPD = (0.61,1.97)), but negatively associated with EAA from DNAmTL clock (posterior mean = −0.13, 95% HPD = (−0.24, −0.04)). Similar results were obtained using the frequentist approach (Supplementary Table 2) as an alternative strategy, with an additional significant finding of a negative association with DunedinPACE (with a small effect size, estimated regression coefficient is −0.03, adjusted $p$-value = 0.01).

## Discussion

In this study of the association between exposure to severe drought in utero and EAA in children, drought exposure associated to multiple clocks. We found a positive association in offspring born to mothers exposed to severe drought during pregnancy based on the Hannum's (first generation) and GrimAge (second generation) clocks, potentially indicating accelerated biological aging in children exposed to severe drought in utero. Additionally, we found a negative association based on DNAmTL. A negative association with DNAmTL is in the expected direction, as shorter telomere length is associated with increased morbidity and mortality. The significance with Hannum is consistent with other studies in children[26], while GrimAge has been found significant in a recent study of in-utero exposure to the Great Depression (based on economic fluctuations)[40]. Results were null for two first generation clocks (Horvath's, SkinBlood), third generation (Dunedin PACE, Dunedin PoAm38), and the two pediatric clocks (PedBe, Wu). The association of in utero drought exposure with DunedinPACE was statistically significant using the frequentist approach, although we approach this finding with caution, as the frequentist method is more susceptible to Type 1 error (see Methods).

Epigenetic clocks are only in their second decade of development, and much remains to be understood about them from a mechanistic perspective[41]. The lack of concordance between clocks tested on the same exposure has been a common feature of epigenetic aging studies. This may partly reflect differences in the way each clock was developed, including participant characteristics of the training sets and the tissues used[10,14].

Additionally, however, biological aging is a multicausal phenomenon, involving accumulated damage, abnormalities, and decline genetically and epigenetically throughout the organism[42–44]. Thus, different epigenetic clocks as well as their DNAm components likely capture distinct mechanisms of biological aging[41]. There are multiple theories about why aging occurs: Two prominent theories point to (a) accumulated damage relevant to tradeoffs that prioritize reproduction over expensive maintenance; and (b) programmatic effects of fitness promoting genes that have deleterious consequences later in life[44]. Either way, consistent with life history theory[45], reproduction is prioritized and the developmental period leading to reproduction is characterized by molecular precision. In adults, EAA indicates relatively more molecular damage (loss of precision) relevant to disease processes that reduce life expectancy, although there isn't agreement concerning EAA's causal centrality in these declines[28]. In children, in contrast, a role in development and/or early life programming has been proposed, although relating these changes to lifetime health is still not well understood[46].

More understanding is needed about the differences between and within epigenetic clocks as they relate to different molecular mechanisms of early life programming, biological aging, chronic diseases, and mortality. A recent study that clustered 5,717 CpGs from fifteen of the best-known clocks into twelve modules offers intriguing insights[41]. The study examined seven out of ten of the clocks tested in our study (DunedinPACE, Wu, and PedBE were not included). Notably, two of the three clocks that were significant in our study (GrimAge and DNAmTL) were found to be similar in composition, together with DunedinPoAm. Since DunedinPACE is a refinement of DunedinPoAm, it is possible that that its composition might also be similar. These three clocks included higher proportions of modules most predictive of mortality risk, two of which also seemed to increase exponentially during development. In contrast, Horvath's, SkinBlood, PhenoAge, and Hannum clocks had higher proportions of modules weakly or inversely predictive of mortality risk – seeming to create a counteracting effect. However, Hannum was distinctive in that it lacked a module present in the other six of these clocks.

As expected in global hot spots for climate change vulnerability, severe drought in our study is characterized by a multitude of stressors

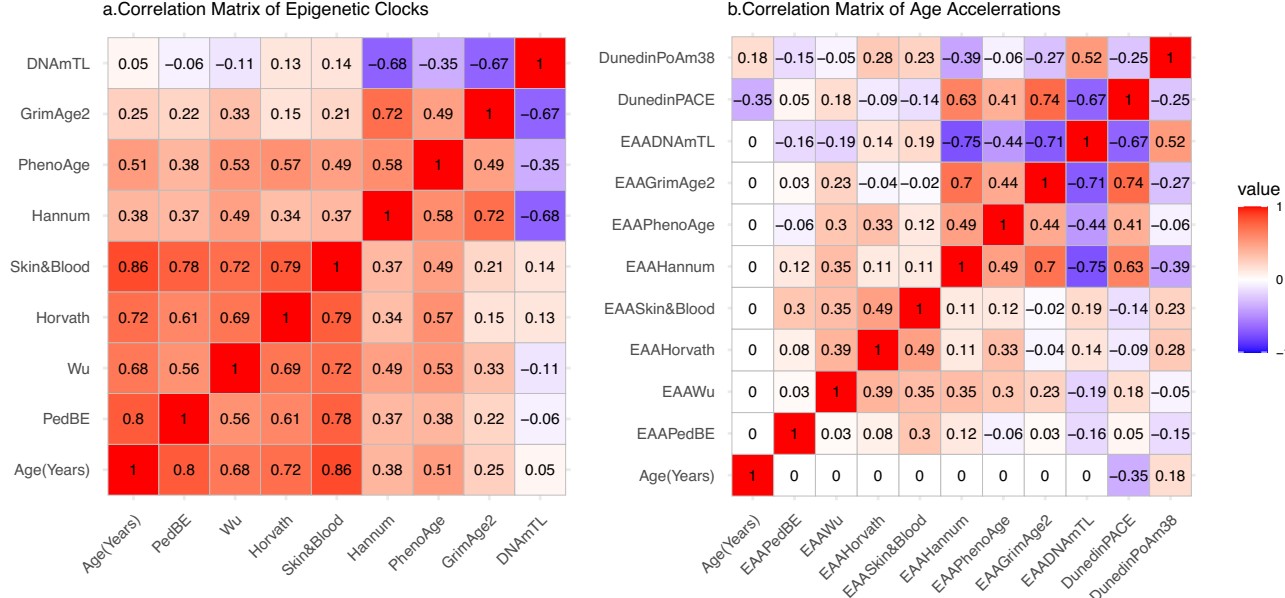

**Fig. 4 | Correlation matrices of epigenetic clocks and epigenetic age accelerations (EAAs).** Figure 4a illustrates the correlation matrix for various epigenetic clocks, while Fig. 4b displays the matrix for correlations between epigenetic age acceleration (EAA) and aging pace metrics. The term "EAAPedBE" refers to the EAA estimated using the PedBE clock, a naming convention consistent across other EAA measurements presented. Positive correlations are denoted by red shades, whereas negative correlations are indicated in blue, with darker shades signifying stronger absolute correlation values. Source data are provided as a Source Data file.

that individuals experienced during their pregnancies[22]. Most frequently reported pregnancy stressors relate to livelihood, food and water insecurity, and intimate partner violence and control. This suggests possibilities for behavioral adaptations that might reduce pregnancy stressors and increase climate resilience. Biologically, pregnant individuals' response to these stressors would be expected to activate the hypothalamic-pituitary-adrenal (HPA) axis. The biological mechanisms for the adaptive response in offspring as it relates to adverse health outcomes and EAA are not as well understood[47], although life history theory predicts a faster reproductive trajectory (also involving the HPA axis) as one possibility[45]. Earlier age at menarche has been found to positively associate to EAA in one study[48], although a subsequent study testing whether earlier age at menarche mediated the association between early life stress and adult EAA had mixed results[47]. Risks for cardiovascular disease and metabolic syndrome have also been found to be associated with earlier age at

menarche, although causation is not established[49]. Some of the components of the clocks that were significant in our study were found strongly associating to cardiovascular outcomes and age at menopause[41]. Notably, we have previously found DNAm mediators of the association between drought and children's body composition near genes relevant to insulin secretion, cardiac function, and cell metabolism, which suggests possible mechanisms for our study's EAA findings[22]. Longitudinal studies are needed to better address the EAA and lifetime health implications of early life exposure to climate stress.

## Methods

### Data collection ethics
All data collection and analysis methods conformed to the principles stated in the Declaration of Helsinki and were approved by Western Michigan University Human Subjects Institutional Review Board [Protocol #17-05-09] and Kenya's National Commission for Science, Technology & Innovation [NACOSTI/P/18/7558/22142; P/19/7558/30004]. All recruitment and informed consent materials were translated and back translated by a multilingual team that included Samburu community partners. The study was explained in the Samburu vernacular at community meetings and to parents and child participants at each data collection visit, with consent (and assent for child participants) obtained at each visit. The study was initiated prior to 2018 IRB rules and is subject to reconsent based on individual subject and Indigenous community agreements.

### Data collection
For this same-sex sibling pair study, we recruited rural Samburu women who had a child exposed in the first trimester of gestation to the peak months of the 2008–2009 drought ($N = 104$), and one or more children of the same sex unexposed to severe drought in utero ($N = 109$). Children's ages (range: 1.81–9.61 years) were documented with vaccination records and birth certificates and calculated to age in years (minimum of two decimal points). Sex of all participating children as identified by parents and children matched sex reported in health records. We conducted qualitative and ethnographically-grounded interviews to elicit reproductive histories, lifetime

**Table 3 | Regression coefficients and 95% highest posterior density (HPD) of drought exposure from the multivariate linear mixed effect model**

|  | Posterior mean | Lower 95% HPD | Upper 95% HPD |
|---|---|---|---|
| EAA$_{PedBE}$[a] | 0.11 | −0.17 | 0.31 |
| EAA$_{Wu}$ | −0.02 | −0.37 | 0.33 |
| EAA$_{Horvath}$ | −0.31 | −0.81 | 0.18 |
| EAA$_{Skin\&Blood}$ | −0.09 | −0.33 | 0.21 |
| **EAA$_{Hannum}$** | **1.34** | **0.74[b]** | **1.96[b]** |
| EAA$_{PhenoAge}$ | 0.78 | −0.59 | 2.15 |
| **EAA$_{GrimAge2}$** | **1.31** | **0.61[b]** | **1.97[b]** |
| **EAA$_{DNAmTl}$** | **−0.12** | **−0.24[b]** | **−0.04[b]** |
| DunedinPACE | −0.03 | −0.13 | 0.07 |
| DunedinPoAm38 | 0.01 | −0.09 | 0.10 |

Bold indicates significant associations. Source data are provided as a source data file.
[a]EAA$_{PedBE}$ denotes the epigenetic age acceleration of PedBE clock.
[b]95% HPD interval not containing zero will be considered as significant association.

stressors, and stressors for each pregnancy[22]. Based on the same-sex sibling design, socioeconomic variables status was the same for drought-exposed and unexposed siblings. The study's epigenetic component was based on children's saliva samples obtained using Oragene-500 kits, as saliva is minimally invasive and appropriate for field conditions.

## DNA methylation

To generate an epigenome-wide dataset of DNAm, we utilized the Illumina MethylationEPIC (EPIC) BeadChip array-based platform. This platform enabled us to gather molecular data comprising over 850,000 methylation marks per individual. To perform the Illumina MethylationEPIC BeadChip array analysis, saliva samples were sent to The University of Michigan's Epigenomics Core for DNA extraction, quality control, and processing. DNA was extracted from saliva using the PureGene Cell and Tissue Kit, according to the protocol suggested for Oragene collection kits (DNA Genotek document PD-PR-00212). Samples were quantified using the Qubit high sensitivity dsDNA assay, and their high molecular weight quality assessed with the TapeStation genomic DNA kit. For each sample, 250 ng were bisulfite converted with Zymo's EZ DNA Methylation kit and using the manufacturer's incubation parameters specific for Illumina MethylationEPIC arrays. Cleaned up samples were then sent to the UM DNA Sequencing core for hybridization to the Infinium MethylationEPIC BeadChip array, washing, and scanning, according to the manufacturer's instructions (Illumina EPIC Datasheet).

## Epigenetic clocks and epigenetic age acceleration

**Epigenetic clocks.** Horvath's pan-tissue clock and Horvath's skin and blood clock are tissue independent clocks and the PedBE is derived from buccal epithelial cells. In contrast, Hannum's single-tissue clock, based on adult blood samples, has higher accuracy for predicting lifespan[27], and accuracy using saliva samples is improved with cell type deconvolution[50,51]. Based on beta-value of DNAm, the PedBE, Horvath, Hannum, skin & blood clocks, PhenoAge, and DunedinPACE were obtained using the R package methylCIPHER from MorganLevineLab[52], and DNAmTl and DunedinPoAm38 were estimated using R package dnaMethyAge from Github[53]. Wu's clock was trained on children's blood samples, and it was estimated by using the coefficients provided in Wu's original paper[38]. GrimAge2 was estimated using the algorithm provided by the authors of DNA methylation GrimAge version 2[54]. More than 80% of the required CpGs are present for all clocks, and missing CpGs are filled with median values from reference dataset[54].

**Epigenetic age acceleration.** From these biological age estimates obtained using the epigenetic clocks, we calculated EAA to measure whether the individuals are biologically younger or older than their chronological age. The PedBE EAA response variable was calculated by using the residuals from a linear model that regresses PedBE clock on the child's chronological age[55]. Similarly, we computed the other EAA response variables using the Wu, Horvath, Hannum, skin & blood, PhenoAge, GrimAge2, and DNAmTL clocks by taking residuals after regressing the corresponding clock on the child's chronological age[10,55]. The DunedinPACE and DunedinPoAm38 directly measure the pace of aging, and therefore could serve as the response variables as EAAs.

## Maternal exposure

Our study focused on first trimester exposure to the 2008–2009 drought as a critical developmental window. We restricted our early gestational drought exposure window to peak months of the drought to capture the highest possible contrast to same-sex siblings not exposed to the 2008–2009 or another severe drought in utero. While this was the most rigorous design to avoid confounding based on early life exposure to the 2008–2009 severe drought, it meant that drought-

exposed children were older than same-sex sibling controls. One child was excluded from the models due to the mother moving outside the drought-catchment area throughout the pregnancy.

## Covariates

*Maternal parity, birth season, sex, cellular heterogeneity, socioeconomic and demographic variables* Gravida indicates the number of pregnancies a woman had prior to the target pregnancy. The child's birth season is a binary variable measuring whether the child was born in the dry or wet season. Child's sex is male or female. Our study is based on whole saliva. We adjusted for cell-type effects, the fractions of a priori known cell subtypes, Epithelial (Epi), Fibroblast (Fib), and Immune cells (ICs, as reference) calculated using the R package, "EpiDISH"[50]. The methods have been described in detail previously[22]. Based on the same-sex sibling pair design, household demographic variables were the same for drought-exposed and unexposed siblings and therefore not included in models.

## Statistical analysis

Due to the same-sex sibling study design and correlation among the eight EAAs (estimated from the PedBE, the Wu, the Horvath, the Hannum, the skin & blood, the PhenoAge, the GrimAge2 and the DNAmTl clocks) and two aging pace response variables (DunedinPACE and DunedinPoAm38), we employed a multivariate linear mixed effect model (MLME) with sibling identifier as random effect to investigate the association between in utero drought exposure and EAA.

We consider the following MLME model:

$$\mathbf{Y_{ij}} = \boldsymbol{\beta_0} + \boldsymbol{\beta_1} T_{ij} + \boldsymbol{\beta_2} \mathbf{X_{ij}} + \mathbf{b_j} + \boldsymbol{\epsilon_{ij}}$$
$$\mathbf{b_j} \sim \mathrm{MVN}(\mathbf{0}, \boldsymbol{\Sigma_b}) \quad\quad (1)$$
$$\boldsymbol{\epsilon_{ij}} \sim \mathrm{MVN}(\mathbf{0}, \boldsymbol{\Sigma_e})$$

where $\mathbf{Y_{ij}} = (Y_{ij,1}, Y_{ij,2}, Y_{ij,3}, Y_{ij,4}, Y_{ij,5}, Y_{ij,6}, Y_{ij,7}, Y_{ij,8}, Y_{ij,9}, Y_{ij,10})^T$ is a 10 by 1 vector of EAA measures and aging pace outcomes of $i$th child of $j$th mother, and $Y_{ij,1}$ denotes the EAA estimated from PedBE clock, $Y_{ij,2}$ from Horvath clock, $Y_{ij,3}$ from Hannum clock, and so on. $T_i$ represents binary exposure of drought, and $\mathbf{X_{ij}}$ is a vector of adjusted covariates: maternal parity, child sex, child birth season, and estimated cell composition variability. In addition, $\boldsymbol{\epsilon_{ij}}$ and $\mathbf{b_j}$ are 10 by 1 vectors of error terms and random intercept, and we assume they are identically and independently distributed (i.i.d.) from multivariate normal (MVN) distribution of mean $\mathbf{0}$ and variance-covariance matrix of $\boldsymbol{\Sigma_e}$ and $\boldsymbol{\Sigma_b}$. Moreover, we assume that $\boldsymbol{\epsilon_{ij}}$ and $\mathbf{b_j}$ are independent. Let $\boldsymbol{\beta_1} = (\beta_{1,1}, \beta_{1,2}, \beta_{1,3}, \beta_{1,4}, \beta_{1,5}, \beta_{1,6}, \beta_{1,7}, \beta_{1,8}, \beta_{1,9}, \beta_{1,10})^T$ be a 10 by 1 coefficient vector representing the association of EAAs and drought exposure, i.e., $\beta_{1,1}$ measures the association of EAA estimated from PedBE clock and early gestational drought exposure. The proposed MLME model in Eq. (1) can be fitted for all EAA measures simultaneously, and it can incorporate not only the correlation of children within mother, but also the association of multiple EAA measures by utilizing the random effects. Since the EAA measures are highly correlated as shown in Fig. 4a, the multivariate modeling approach is more powerful than univariate approach[56].

Moreover, the MLME provides a solution to the multiplicity issue by summarizing simultaneously all EAA measures of interest instead of fitting many separate univariate linear mixed effects models for each EAA. For frequentist approach, a maximum likelihood (ML) approach based on the joint marginal likelihood of EAA measures can be used for estimating the fixed and random effects parameters in MLME model. However, the ML approach requires numerical integration techniques with respect to the random effects and the large number of parameters included in the models. To overcome the computational burden, we utilize the Bayesian approach in conjunction with Markov chain Monte Carto (MCMC) methods, i.e., Gibbs sampling, to obtain the posterior

distribution of the parameters of interest for parameter estimation and inference. We use the standard conjugate prior distribution for the parameters of interest, that is non-informative multivariate normal distribution for regression coefficients in (1), i.e., $\boldsymbol{\beta_1} \sim \text{MVN}(\mathbf{0}, 10^9 \mathbf{I_{10}})$, where $\mathbf{I_{10}}$ is the 10 by 10 identity matrix, and inverse-Wishart distribution for variance and covariance matrix of random effect $\mathbf{b_j}$ and error terms $\boldsymbol{\epsilon_{ij}}, \boldsymbol{\Sigma_b}, \boldsymbol{\Sigma_e} \sim \text{IW}(10, \mathbf{I_{10}})$. Under this setting, we assumed the independent prior information on $\boldsymbol{\beta_1}$, $\mathbf{b_j}$, $\boldsymbol{\Sigma_b}$ and $\boldsymbol{\Sigma_e}$. The number of interactions of Gibbs sampling was 20,000 with the first 5000 samples discarded as burn in. Using the R package MCMCglmm, we obtained the marginal posterior distribution, and computed the posterior mean and 95% highest posterior density (HPD) interval of regression coefficients[57]. The convergence in MCMC for each parameter in the MLME model was further inspected using the trace plots. Our statistical analyses were performed using R Version 4.1.2, and we used 0.05 to be a significance threshold.

## Strengths and limitations

Our study contributes to ethnic diversity of EAA research and adds much needed climate exposure findings. Also, it contributes findings from a population engaged in a climate-vulnerable livelihood, based on findings from an underrepresented ethnic group in northern Kenya living in a global hot spot for climate change vulnerability. Our same-sex sibling study design is an additional strength. Also, by analyzing the EAA response variables simultaneously using a multivariate approach, we account for the correlations between EAA measures, thus decreasing the likelihood of a Type 1 error without requiring any additional multiple testing adjustment. Limitations of our study include the fact that drought-exposed siblings are older than unexposed siblings. However, the EAA measures do not associate significantly to chronological age, and all children (both drought-exposed and unexposed) were under ten years old and pre-pubertal based on observed Tanner stage. Another limitation is that it is possible that our models are biased towards the null for some clocks, based on differences between participant characteristics of our study sample and those of participants used to develop the clocks[24]. Watkins and colleagues have recently pointed out a lack of attention given to participant characteristics for samples on which clocks are based in spite of the recognized problem raised by a tendency to use unrepresentative sociodemographic samples. These authors suggest that researchers working with existing clocks should compare participant characteristics of their study to characteristics of participants used in the development of the clocks to the extent that information is available[24]. We have described available information for each clock in the main text for this reason.

## Reporting summary

Further information on research design is available in the Nature Portfolio Reporting Summary linked to this article.

## Data availability

The source data for all Figures and Tables, including data used to support the findings of this study are supplied with this paper in the Source Data file. Rainfall variables for Fig. 1 were calculated from high resolution publicly available historical data (CHIRPS) on the Famine Early Warning Systems Network (FEWSNET) https://earlywarning.usgs.gov/fews). DNA methylation data are based on the Illumina MethylationEPIC (EPIC) BeadChip array-based platform. The individual-level DNA methylation data and the CHIRPS individual-level location and pregnancy-timed data (used for illustration purposes in Fig. 1, not for data analyses) are available under restricted access due to privacy and ethical restrictions because the research partners of this study are a vulnerable group of African Indigenous peoples. Access can be obtained by contacting the corresponding author (bilinda.straight@wmich.edu) as follows: Queries for access to verify results will

receive a response within 2 weeks, and access will be granted as immediately as possible after approval from the Western Michigan University IRB, but in no more than 2 weeks from approval. Access for new studies will receive a response within 2 weeks and will be subject to restrictions imposed by Kenya's National Commission for Science, Technology, and Innovation (NACOSTI) and the Indigenous community. • Epigenetic clocks were derived using R package *methylCIPHER* from MorganLevineLab, R package *dnaMethyAge* from Github, R function derived from original clock paper, and R code provided by the authors of "DNA methylation GrimAge version 2". • Cell type proportions were estimated using R package *EpiDISH*. • Figures 1 and 3 were generated using Microsoft Excel for Mac Version 16.83. • Figure 4 was generated using R packages *ggcorrplot* and *patchwork*. • All code is run in R version 4.1.2. • Statistical analyses were conducted using the following packages: *MCMCglmm*, *lmerTest* and *lme4*. Source data are provided with this paper.

## Code availability

R Code used for analyses can be found at: https://github.com/DuyNgoStats/BayesianEAA. and at the following https://doi.org/10.5281/zenodo.10854895. Code Detailed Description. *Step 1: Missing Values and Imputation*. • Check methylation sites missing value percentage and fill the missing value with median value from the reference dataset. *Step 2: Epigenetic Clock and Age Acceleration Estimation*. • Calculate Epigenetic clocks using R package methylCIPHER and dnaMethyAge. • Regress each clock on chronological age and take the corresponding residual as the Epigenetic age acceleration measure. *Step 3: Cell Type Estimation*. • Estimate the cell type proportion (Epithelial, Fibroblast, with Immune cells as reference) with beta values of CpG sites using R package EpiDISH. *Step 4: Multivariate Linear Mixed Effect Model (MLME) and MCMC Diagnostics*. • Set up prior distributions for MLME model parameters. • For the MLME model, age acceleration measures serve as the multivariate dependent variables, with drought as the exposure variable and cell type, maternal parity, child's sex, and child's birth season as covariates. • Inspect MCMC samples with trace plots and density plots.

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

## Acknowledgements

The drought study was funded by National Science Foundation (Award # 1728743, BS) and Western Michigan University Faculty Research and Creative Activities Award (62-2017, BS). Any opinions, findings, and conclusions or recommendations expressed in this material are those of the author(s) and do not necessarily reflect the views of the National Science Foundation. We are grateful to Kenya's National Commission for Science, Technology and Innovation (NACOSTI) and the Samburu County government for permission to conduct this research. We are also grateful to our Samburu research partners, research assistants, and their communities, who have welcomed us into their homes and been a pleasure to work with. Finally, we would like to thank the Horvath group for providing an R script to estimate GrimAge.

## Author contributions

X.Q. calculated DNAm age and EAA, performed statistical modeling, and co-wrote the paper draft; B.S. conceived the study concept, co-conceived the design, led all phases and components of the study from recruitment through analysis, and co-wrote the paper draft; D.N. mentored X.Q. on DNAm modeling and contributed to paper drafts and revisions; C.E.H. supervised all data collection with participant children and contributed to paper revisions; A.N. consulted on psychological instrumentation and contributed to methods and implementation for quantifying qualitative data; C.O.O. contributed to guidance and support in the field, expertise on cultural anthropological components of the study, and contributed to paper revisions; C.L. performed and supervised DNA extraction and QC and DNAm laboratory analyses and QC; B.L.N. contributed to study design, expertise at all phases of the study, and contributed to paper revisions.

## Competing interests

The authors declare no competing interests.
