## [Peer Review File · Nature Communications]

Severe drought exposure in utero associates to children's epigenetic age acceleration in a global climate change hot spotEditorial Note: This manuscript has been previously reviewed at another journal that is not operating a transparent peer review scheme. This document only contains reviewer comments and rebuttal letters for versions considered at Nature Communications.

Reviewers' Comments:

Reviewer #2:

Remarks to the Author:

This study investigates the association between in-utero exposure to drought and epigenetic age acceleration (EAA) in children. EAA was assessed using 8 publicly available epigenetic clocks and 2 pace-of-aging estimators. The study relied on a rather small sample of 104 same-sibling pairs (213 children) from northern Kenya, where populations would be especially vulnerable to climate change. The limitation due to the small sample size may be partly mitigated by the paired design.

The manuscript is clear and well-written. The authors answered satisfactorily to my previous questions and the quality of the manuscript and figures was improved.

I am still skeptical regarding the reliability of the observations and conclusions that can be drawn from so-called epigenetic clocks, due to the lack of validation against ground truth or any gold standard measurement for "biological aging". Moreover, in my opinion, there is no clear definition of what is measured by "epigenetic aging". I think this was an important point to be argued; which has been taken into consideration in the revised manuscript.

That said, the process adopted for computing EAA matches the current standards. I appreciate the efforts and methodological developments which were made to enrich the analysis and interpretation of these data. Importantly, the authors highlighted and discussed some very interesting points and thoughts in the discussion section.

Overall, I believe this work is a valuable and original add-on to the field.

Reviewer #3:

Remarks to the Author:

The manuscript, "Drought and heat exposure in utero associate to children's epigenetic age acceleration in a global climate change hot spot," addresses an important topic, which reinforces the complexity of early life exposure to climate change (severe drought or high ambient temperatures) and epigenetic aging relationships to lay a foundation for future studies investigating climate change and longevity. The paper examines drought exposure in utero and early childhood exposure to ambient heat with eight epigenetic clocks (Horvath, Hannum, Skin&Blood, PedBE, Wu, PhenoAge, GrimAge, and DNAmTL) in saliva samples from 213 children (104 drought-exposed children and 109 same-sex sibling controls in northern Kenya). Epigenetic aging was measured in saliva with the Illumina MethylationEPIC BeadChip array. Pace of aging measures were added and include DunedinPACE and DunedinPaAm38. The authors state that this study contributes to ethnic diversity of epigenetic aging studies and adds much needed data to the climate change exposure literature. I would also add that additional studies in children is also a strength as it is still unclear what the direction of epigenetic aging means in children. This is especially important as there have been counterintuitive findings reported in previous studies of children stress and epigenetic aging. Comments made in feedback from previous versions of this manuscript were sufficiently addressed. The additional details on the different generations of epigenetic clocks are very helpful. I appreciate

the inclusion of how the information was gathered from the mothers in their own language. The addition of Figure 1 greatly clarifies the rainfall comparison between the drought-exposed (n=104) vs. the same-sex drought-unexposed sibling (n=109).

Some specific comments:

There is an empty table before the materials and methods section.

References are listed twice and are different from each other. I believe that the second list is more accurate as you cite references 58-80 and the first list only goes to 57. Please fix.

Reviewer #4:

Remarks to the Author:

This is a well-written manuscript with elegantly conducted analyses. The revisions provided nicely enhance the results and impact of the study. In general, the addition of pediatric clocks is an important improvement as are the additional considerations of potential residual confounding not previously considered. In all, a nice responsive resubmission.

Response to Reviewers

We are very grateful to the reviewers for their time and careful attention to our paper. Reviewers 1 and 4 did not make suggestions for further revision. Reviewer 3 noted an empty table and that references were listed twice. Our newly uploaded manuscript version does not have empty tables and eliminates redundancy in the references cited.